# Determinants of species-specific utilization of ACE2 by human and animal coronaviruses

Qingxing Wang[1], Sabrina Noettger[1], Qinya Xie[1], Chiara Pastorio[1], Alina Seidel[1], Janis A. Müller [1,5], Christoph Jung[2,3,4], Timo Jacob [2,3,4], Konstantin M. J. Sparrer [1], Fabian Zech[1] & Frank Kirchhoff [1✉]

Utilization of human ACE2 allowed several bat coronaviruses (CoVs), including the causative agent of COVID-19, to infect humans directly or via intermediate hosts. However, the determinants of species-specific differences in ACE2 usage and the frequency of the ability of animal CoVs to use human ACE2 are poorly understood. Here we applied VSV pseudoviruses to analyze the ability of Spike proteins from 26 human or animal CoVs to use ACE2 receptors across nine reservoir, potential intermediate and human hosts. We show that SARS-CoV-2 Omicron variants evolved towards more efficient ACE2 usage but mutation of R493Q in BA.4/5 and XBB Spike proteins disrupts utilization of ACE2 from Greater horseshoe bats. Variations in ACE2 residues 31, 41 and 354 govern species-specific differences in usage by coronaviral Spike proteins. Mutation of T403R allows the RaTG13 bat CoV Spike to efficiently use all ACE2 orthologs for viral entry. Sera from COVID-19 vaccinated individuals neutralize the Spike proteins of various bat *Sarbecovirus*es. Our results define determinants of ACE2 receptor usage of diverse CoVs and suggest that COVID-19 vaccination may protect against future zoonoses of bat coronaviruses.

---

[1] Institute of Molecular Virology, Ulm University Medical Center, 89081 Ulm, Germany. [2] Institute of Electrochemistry, Ulm University, 89081 Ulm, Germany. [3] Helmholtz-Institute Ulm (HIU) Electrochemical Energy Storage, 89081 Ulm, Germany. [4] Karlsruhe Institute of Technology (KIT), 76021 Karlsruhe, Germany. [5] Present address: Institute of Virology,  Philipps University Marburg, 35043 Marburg, Germany. ✉email: Frank.Kirchhoff@uni-ulm.de

Coronaviruses (CoVs) have been detected in many animal species, including bats, swine, cattle, horses, camels, cats, raccoon dogs, rodents, rabbits, ferrets, civets and pangolins[1,2]. They are well known to cross-species barriers and have been successfully transmitted to humans at least seven times[3]. Bats are the reservoir hosts and presumably all human CoVs (hCoVs) originate from bat viruses, although transmission via intermediate hosts frequently facilitated viral zoonoses[2,4]. Four endemic human coronaviruses (hCoV-229E, -OC43, -NL63, -HKU1) have been circulating in the human population for at least several decades and are responsible for a significant proportion of seasonal common colds[5,6]. While they are meanwhile well adapted to humans, it has been suggested that they may have been more pathogenic early after transmission to humans[7]. Three additional coronaviruses that emerged from viral zoonoses in the last 20 years cause severe disease. SARS-CoV-1 was identified in 2003 as the causative agent of severe acute respiratory syndrome (SARS), infected ~8.000 people, and was associated with a mortality of ~10%[8]. The Middle East Respiratory Syndrome-CoV (MERS-CoV) emerged in 2012 with case fatality rates of almost 40%[9]. Only 7 years later (December 2019), SARS-CoV-2, the causative agent of the COVID-19 pandemic, was first detected[10,11]. SARS-CoV-2 has a case fatality rate of less than 1% but spread at an alarming rate and has caused over 700 million infections worldwide as of March 2023. Both SARS-CoV-1 and SARS-CoV-2 belong to the *Sarbecovirus* subgenus of β-coronaviruses, which are mainly found in bats[12] but have also been detected in pangolins and civets[13–17].

Due to effective vaccines and increasing immunity, we are just gaining control over the SARS-CoV-2 pandemic. However, at least seven independent zoonotic transmissions of animal coronaviruses, including three highly pathogenic ones within the past 20 years, clearly highlight that future zoonoses of bat CoVs pose a significant threat. The Spike (S) proteins of three of the human coronaviruses, SARS-CoV-1, SARS-CoV-2, and HCoV-NL63, utilize the angiotensin-converting enzyme 2 (ACE2) receptor for infection of human target cells[18–20]. Thus, the ability of CoVs to use human ACE2 for efficient entry plays a key role in successful zoonotic transmission. Recently, a variety of bat CoVs that are related to human SARS- and MERS-CoVs have been discovered and isolated[21,22]. Alarmingly, a new study indicates frequent spillover of diverse bat CoVs in human communities in contact with wildlife[23]. Previous studies provided some insight into the ability of Spike proteins from SARS-CoV-1, early SARS-CoV-2 variants, and the closely related bat CoV RaTG13 to interact with different ACE2 orthologs[13,24–29]. Altogether, however, the ability of human, bat and other animal CoVs to utilize the ACE2 receptors of reservoir bats species, putative intermediate hosts, and humans is poorly understood. Therefore, we analyzed the ability of 26 Spike proteins from a variety of bat, civet, pangolin and human coronaviruses to use ACE2 receptors from bat reservoir species, potential intermediate civet, pangolin, racoon dog, camel, ferret and pig hosts, as well as humans. We found that single amino acid changes in Spike proteins of human and bat CoVs can drastically change their ability to utilize ACE2 receptors from these different species. Finally, we show that sera from individuals vaccinated against SARS-CoV-2 neutralize VSV pseudoparticle infection mediated by Spike proteins from highly divergent bat CoVs but were poorly active against Omicron XBB.1 and XBB.1.5 Spikes.

## Results

### Expression of viral Spike proteins and ACE2 receptors from different species.
Bats are considered the original reservoir hosts of all zoonotic coronaviruses. In several cases, however, transmission to humans likely involved intermediate hosts, such as palm civets (SARS-CoV-1)[17], pangolins or raccoon dogs (SARS-CoV-2)[15,16,30], dromedary camels (MERS-CoV, hCoV-229E)[31], pigs (hCoV-OC43) or mice (hCoV-HKU-1) (Fig. 1a). To better assess the risk of future transmissions of coronaviruses to humans, we generated a collection of expression constructs of S proteins from divergent bat CoVs, close relatives of SARS-CoV-1 and SARS-CoV-2 found in civets and pangolins, respectively, as well as all seven hCoVs (Fig. 1b and Supplementary Table 1). To analyze their function, we pseudotyped vesicular stomatitis virus (VSV) particles lacking the glycoprotein (G) gene but encoding the green fluorescent reporter protein (VSVΔG-GFPpp) with these S proteins. Western blot analyses showed that all 26 S proteins were expressed at detectable albeit variable levels in transfected HEK293T cells (Supplementary Fig. 1a). S-mediated entry depends on proteolytic processing, e.g., by furin at the S1/S2 site, as well as TMPRSS2 or cathepsins[32,33]. All S proteins were processed and detected in the culture supernatants indicating incorporation into VSVΔG-GFPpp (Supplementary Fig. 1a).

To determine species-dependent differences in the ability of ACE2 to mediate entry of human and animal CoVs, we generated expression constructs for ACE2 orthologues from Greater horseshoe bats (*Rhinolophus ferrumequinum, Rf*), Intermediate horseshoe bats (*Rhinolophus affinis, Ra*), Sunda pangolin (*Manis javanica*), Masked palm civet (*Paguma larvata*), Common raccoon dog (*Nyctereutes procyonoides*), Ferret (*Mustela putorius furo*), Camel (*Camelus dromedaries*), Pig (*Sus scrofa domesticus*) and humans (Fig. 1c and Supplementary Table 2). ACE2 receptors from all species were expressed at similar levels in transfected HEK293T cells (Supplementary Fig. 1b). This collection of expression constructs allowed us to analyze the ability of S proteins from diverse animal and human CoVs to utilize ACE2 from reservoir and intermediate hosts for viral entry.

### Species-specific ACE2 usage by SARS-CoV-2 variants.
Given the enormous spread of SARS-CoV-2, it is a concern that emerging variants might establish reservoirs in animal hosts, such as ferrets or pigs, mutate and be transmitted back to humans[34–36]. To better assess this, we analyzed the ability of Spike proteins of different SARS-CoV-2 variants including Omicron variants of concern (VOCs) to exploit ACE2 from various animal species. Infection assays using VSVΔG-GFPpp showed that S proteins of the SARS-CoV-2 Hu-1, Delta and Omicron BA.1, BA.2, BA.4/5, XBB.1, and XBB.1.5 variants mediated infection of HEK293T cells expressing ACE2 receptors derived from civet, pangolin, racoon dog, camel, ferret, pig and intermediate horseshoe (*Ra*) bats (Fig. 2a). Notably, cells expressing pig ACE2 often showed the highest infection efficiency. In comparison, the S proteins of the early Hu-1 variant, the long-dominating BA.5 VOC (identical to BA.4), and the currently widespread XBB.1.5 VOC were unable to utilize ACE2 from greater horseshoe bats (*Rf*) (Fig. 2a). To challenge these findings and to assess the infection kinetics, we used an approach allowing automated quantification of the number of VSVpp-infected (GFP +) cells over time[37]. The results confirmed that the BA.5, XBB.1 and XBB.1.5 S proteins usually show increased infection efficiencies compared to all remaining SARS-CoV-2 variants[38] but are unable to use *Rf* ACE2 for infection (Supplementary Fig. 2). To further determine the fusogenic activity of SARS-CoV-2 S proteins in cells expressing human or bat ACE2, we performed quantitative cell–cell fusion assays. All SARS-CoV-2 S proteins promoted the formation of large syncytia in cells expressing human or *Ra* ACE2 (Supplementary Fig. 3). In agreement with the VSVpp infection data, however, the Hu-1 as well as BA.5, XBB.1 and XBB.1.5 S proteins

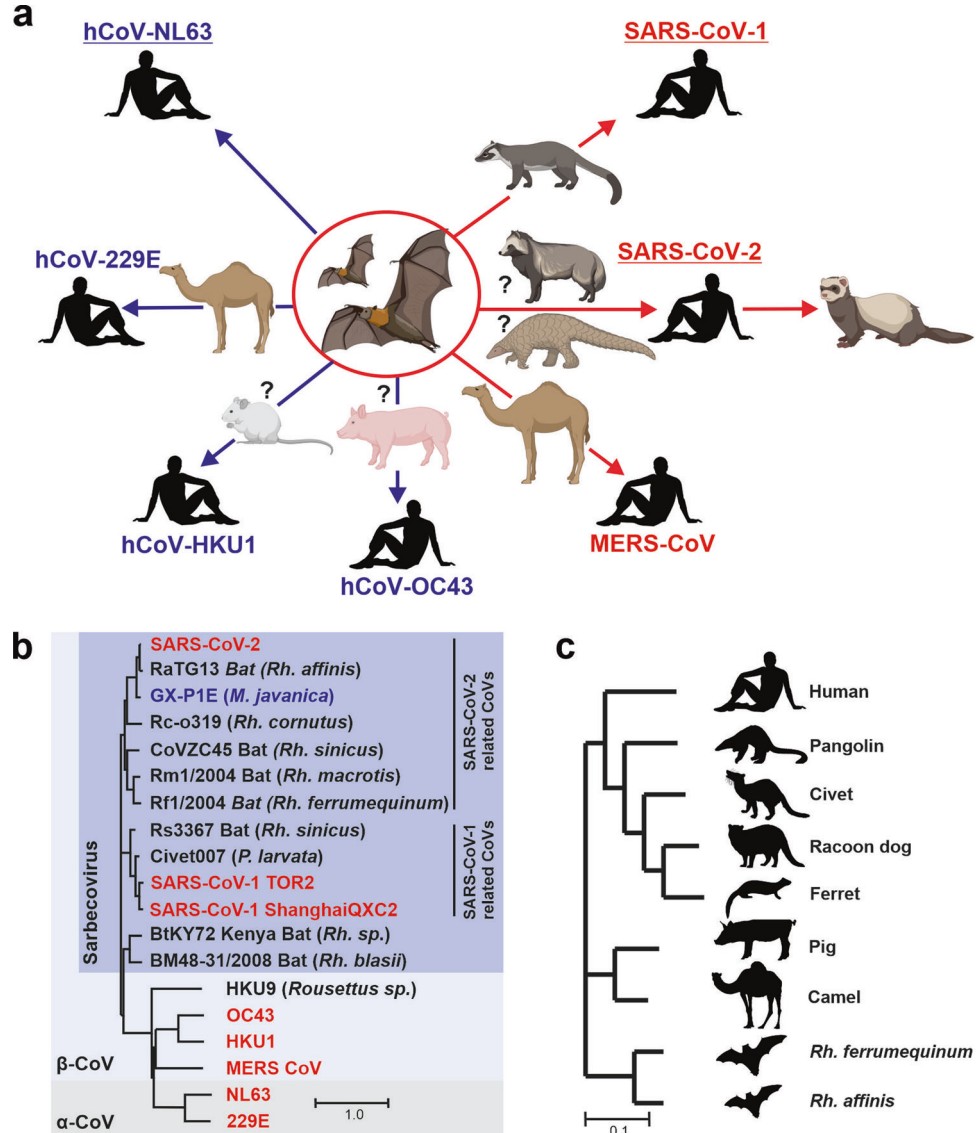

**Fig. 1 Potential origin of human CoVs and phylogenetic relationship of Spike and ACE2 proteins analyzed. a** Schematic representation of the potential cross-species transmissions that led to the emergence of the seven human CoVs. Highly pathogenic CoVs are indicated in red, and circulating CoVs in blue. Those utilizing ACE2 are underlined. In some cases, the intermediate hosts are unknown or under debate. Images were derived from Biorender. **b**, **c** Phylogenetic relationship between the Spike (**b**) and ACE2 (**c**) amino acid sequences from the indicated viral strains or species, respectively.

showed little if any activity in mediating membrane fusion in cells expressing *Rf* ACE2. Thus, Omicron VOCs evolved an increasing ability to use ACE2 orthologs of most species during adaptation to humans, but the BA.5, XBB.1, and XBB.1.5 variants lost the ability to use *Rf* ACE2 for entry.

**Residues R493 in Spike and D31/H41 in *Rf* ACE2 affect BA.5 entry**. The S protein of the Omicron BA.5 variant differs only by deletion of amino acids 69 and 70 and changes of L452R, F486V, and R493Q from its BA.2 precursor[39]. To determine which of these mutations is responsible for the loss of *Rf* ACE2 usage, we introduced them individually into the BA.2 S protein. All BA.2 mutant S proteins were efficiently expressed and processed (Fig. 2b). Mutation of R493Q disrupted the ability of the BA.2 S to use *Rf* ACE2 for viral entry (Fig. 2c). It has been reported that reversion of R493Q (Q493 is found in early SARS-CoV-2 strains, including Hu-1) restores affinity for human ACE2 and consequently the infectiousness of BA.4/5[40]. Based on the modeled structure, R493 in S interacts with D31 and E35 in the *Rf* ACE2

and these interactions are disrupted by the R493Q substitution (Fig. 2d). To further define S-ACE2 interactions, we substituted nine amino acids of the *Rf* ACE2 by those found in *Ra* ACE2 (Supplementary Fig. 4). All mutant ACE2 proteins were expressed at similar levels but mutations of D31N and H41Y as well as (to a lesser extent) K27I and N38D allowed them to mediate infection via the BA.5 S protein (Fig. 2e). Our results demonstrate that the R493Q change in CoV-2 S disrupts utilization of ACE2 from *Rf* bats that are widespread in Europe, Northern Africa, and Asia. In addition, we show that amino acid residues 31 and 41 play a role in the ability of ACE2 to serve as a receptor for CoV infection.

**Species-specific ACE2 usage by animal relatives of SARS-CoVs**. To determine the species-specificity of ACE2 receptor usage by animal CoVs related to human SARS-CoVs, we overexpressed human or animal-derived ACE2 in HEK293T cells and examined their susceptibility to VSVpp infection mediated by S proteins from bat, pangolin, civet and human CoVs. The collection

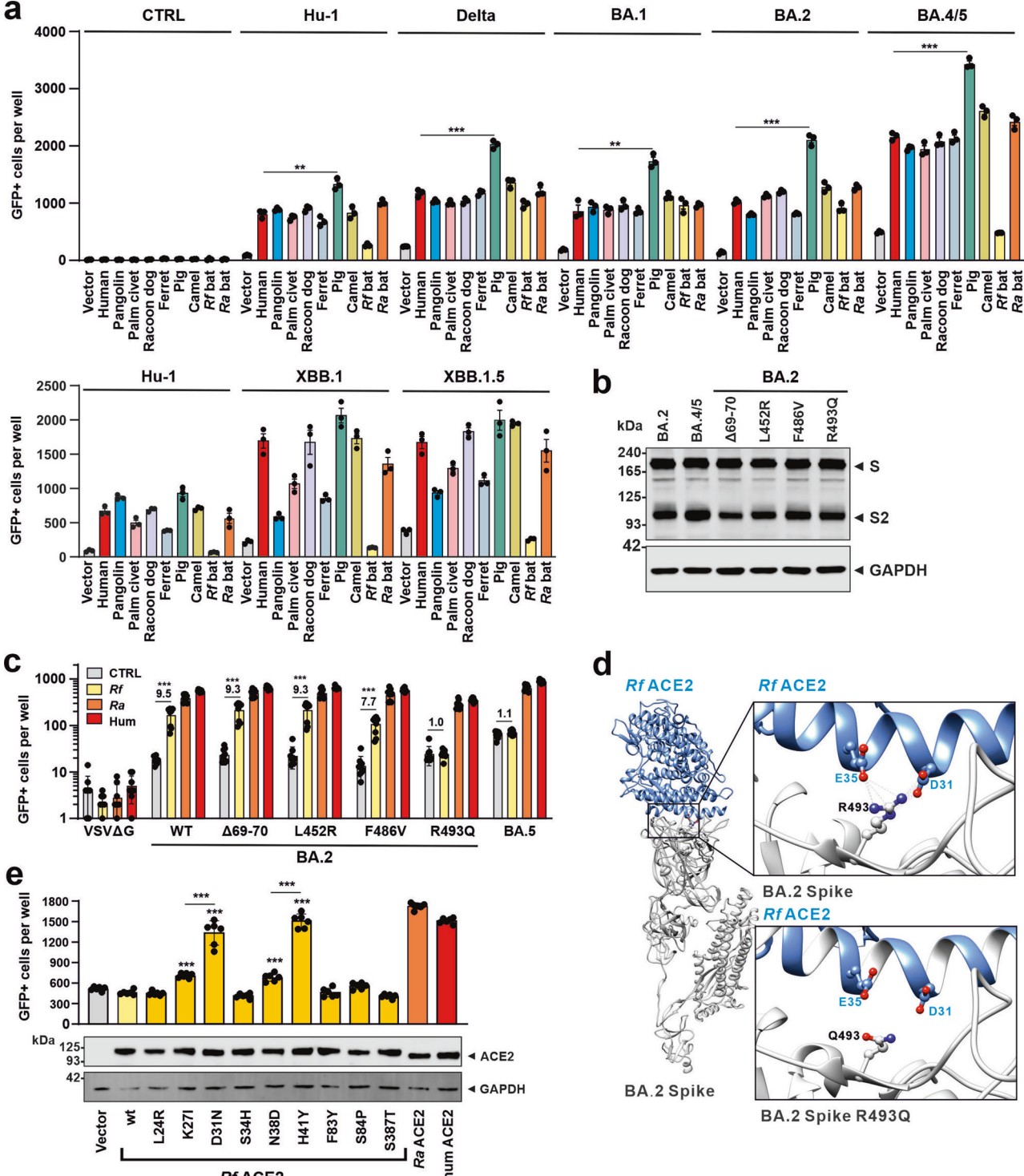

**Fig. 2 Utilization of ACE2 receptors from different species by SARS-CoV-2 variants. a** Automatic quantification of infection events of HEK293T cells expressing the various ACE2 receptors and transduced with VSVΔG-GFP pseudotyped with the indicated SARS-CoV-2 S proteins or lacking S (CTRL). Bars in all panels represent the mean of three experiments (± SEM). Statistical significance was determined by unpaired *t* tested; *$P < 0.05$; **$P < 0.01$; ***$P < 0.001$. Background was generally increased for BA.4/5, XBB.1 and XBB.1.5 S, possibly due to low levels of ACE2-independent infection of the target cells. **b** Immunoblot of whole cells lysates of HEK293T cells expressing BA.2, BA.4/5 or the indicated mutant S proteins. Blots were stained with anti-V5 tag and anti-GAPDH. **c** Infection of HEK293T cells transfected with an empty control vector (gray) of the *Rf* (yellow), *Ra* (orange), or human (red) ACE2 receptors. Numbers above bars indicate *n*-fold enhancement compared to control. Bars in all panels represent the mean of three experiments (± SEM). **d** Schematic diagram of wild-type and R493Q BA.2 S with the *Rf* ACE2. Potential van der Waals interactions and hydrogen bonds of BA.2 spike R493 with *Rf* ACE2 D31 and E35 are indicated by dash back lines and pink lines, respectively. **e** Ability of wild-type and mutant *Rf* ACE2 receptors for BA.5 S-mediated infection of VSVpp. Bars in all panels represent the mean of three experiments (± SEM). If not indicated otherwise, *P* values indicate difference to the wild-type *Rf* ACE2 receptor. Human and *Ra* ACE2 are shown for control.

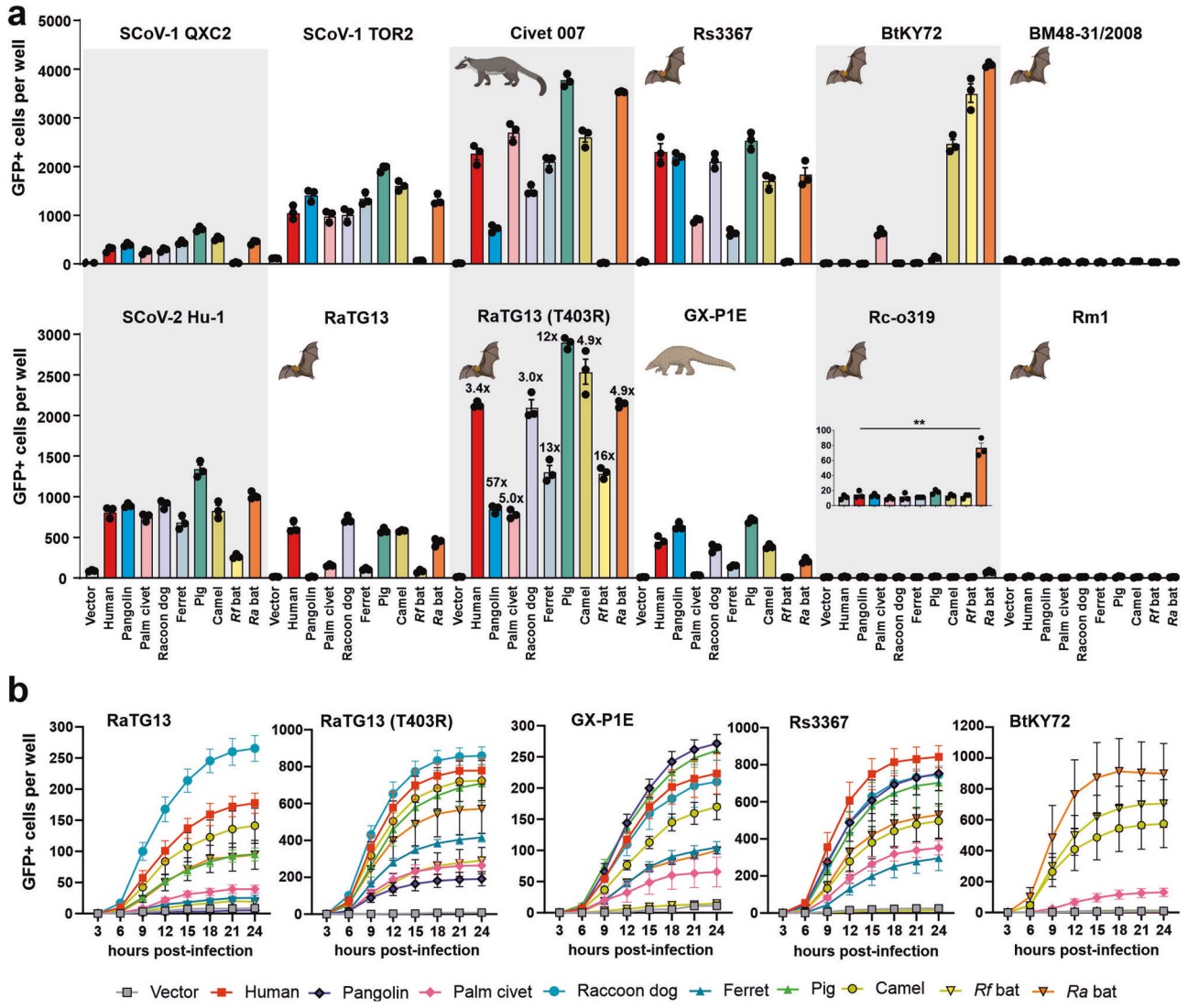

**Fig. 3 Species-specific utilization of ACE2 by S-proteins of bat, pangolin, and civet CoVs. a** Automatic quantification of infection events of HEK293T cells expressing the indicated ACE2 receptors infected by VSVpp carrying S proteins of SARS-CoVs or animal CoVs. Bars represent the mean of three experiments (± SEM). Statistical significance was tested by unpaired $t$ test; **$P < 0.01$. **b** Infection kinetics of ACE2 expressing HEK293T cells infected by VSVpp containing the indicated mutant S proteins. Infected GFP+ cells were automatically quantified over a period of 24 h. Bars in all panels represent the mean of four experiments (± SEM).

encompassed S proteins of the closest relatives of SARS-CoV-1 and SARS-CoV-2 detected in bats and potential intermediate civet and pangolin hosts, as well as some more distantly related bat CoVs (Fig. 1b). The early SARS-CoV-1 QXC2 strain used all ACE2 orthologs (except *Rf*), albeit generally with low efficiency (Fig. 3a). In comparison, the closely related TOR2 variant isolated in 2003 from a patient with SARS in Toronto[41], showed about three- to fourfold higher infection rates indicating acquisition of increased ACE2 affinity during spread in humans (Fig. 3a). Surprisingly, VSVpp containing S proteins from closely related CoVs isolated from civets (98.1% identity)[42] and Rs3367 isolated in 2011 or 2012 from a Chinese Rufous Horseshoe Bat (*Rhinolophus sinicus*) (91.9% identity)[13] infected cells expressing all ACE2 receptors (except *Rf*) including the human ortholog with higher efficiency than those carrying SARS-CoV-1 S (Fig. 3a).

The S protein of BtKY72 obtained from Kenyan *Rhinolophus* bats[43] efficiently utilized the ACE2 receptors of *Rf* and *Ra* bats, as well as (unexpectedly) camel ACE2 and (to a lesser extent) palm civet ACE2 for infection (Fig. 3a). In contrast, the S protein of the related BM48 bat CoV[44] (Fig. 1b) did not use ACE2 for entry.

Sequence alignments show that all four ACE2 orthologs allowing BtKY72 S-mediated infection contain mutations in an otherwise conserved lysin residue K31N/D/E/T (Supplementary Fig. 5a). Amino acid 31 in ACE2 is close to the receptor-binding site (RBD) of S proteins (Supplementary Fig. 5b) and substitution of D31N in *Rf* ACE2 allowed it to serve as entry receptor of BA.5 S (Fig. 2e). Usage of *Ra* ACE2 for infection agrees with the previous finding that it interacts with the RBD of the BtKY72 bat CoV[24]. We found that *Rf* ACE2, which can be used far less than *Ra* ACE2 as entry receptor, also allows efficient infection by the BtKY72 S (Fig. 3a). The geographical distribution of *Ra* and *Rf* bats overlap and comprises North Africa and large parts of Europe and Asia. Altogether, these results further support that residue 31 in ACE2 plays an important role in S interaction and suggest that some bat CoVs have the potential to spread across three continents.

Spike proteins of the closest non-human relatives of SARS-CoV-2, RaTG13 sampled from a *Rhinolophus affinis* horseshoe bat in 2013 in Mojiang, Yunnan (China) that shows 96.1% sequence identity to SARS-CoV-2[10], and GX-P1E obtained from tissue samples collected from Malayan pangolins in 2017 and

showing ~85.3% sequence identity[15], used human ACE2 almost as efficiently as the early Hu-1 CoV-2 strain (Fig. 3a). However, both were more restricted than Hu-1 S in the usage of ACE2 receptors from other species. For example, the wild-type RaTG13 S allowed little if any infection of cells expressing pangolin, palm civet, ferret and *Rf* ACE2. We have previously shown that a single change of T403 to R (found in the S proteins of most other *Ra* bat CoV S) in RaTG13 S strongly enhances infection via human ACE2[26]. Thus, we examined whether it has broader impact on ACE2 usage. Strikingly, utilization of all ACE2 orthologues was increased by 3- to 57-fold and the mutant T403R RaTG13 S was capable of using all ACE2 orthologs at least as efficiently as the SARS-CoV-2 Hu-1 S for infection (Fig. 3a). Sequence analyses showed that E37 in ACE2 that is critical for the enhancing effect of the T403R mutation is conserved in all but one ACE2 ortholog analyzed (Supplementary Fig. 6). The exception was the civet ACE2, which contains a Q at position 37 and showed relatively poor efficiency in mediating infection by the T403R RaTG13 S (Fig. 3a).

The S proteins of bat CoVs that are more distantly related to SARS-CoV-2, i.e., Rc-o319, Rm1 and Rf1 detected in *R. biasii*, *R. macrotis* and *Rf* bats, respectively (Fig. 1b), were generally unable to use any ACE2 ortholog for infection (Fig. 3a). The only exception was that the Rc-o319 S mediated infection via *Ra* ACE2, with significant albeit marginal efficiency (Fig. 3a). Automated quantification of the number of VSVpp-infected cells over time confirmed that the bat Rs3367 and pangolin GX-P1E Spikes utilize all but the *Rf* ACE2 receptor with similar kinetics but varying efficiencies (Fig. 3b). The results also confirmed that T403R in RaTG13 S strongly enhances infection by all nine ACE2 orthologs (Fig. 3b). Finally, the infection kinetics verified that the BtKY72 S uses camel, *Ra* and *Rf* ACE2 and (less efficiently) civet but not human, pangolin, racoon dog, ferret or pig ACE2 for infection (Fig. 3b). Although the BtKY72 S efficiently utilized *Rf* and *Ra* ACE2 for infection of VSVpp (Fig. 3a) it did not induce syncitia formation in cell-to-cell fusion assays (Supplementary Fig. 7). Altogether, S proteins from bat CoVs show striking and often species-specific differences in their ability to use ACE2 for infection, which are in part determined by amino acid variations at position 403 in the viral S protein and 31 in the ACE2 receptors.

**G354 in ACE2 is critical for infection by hCoV-NL63.** Of the remaining five human CoVs only the NL63 S is known to use human ACE2 for infection[19]. We addressed the possibility that S proteins of hCoV-MERS, -229E, -OC43, -HKU1 or HKU9 from the related *Rousettus* bats may use ACE2 orthologs from other species. In agreement with published data[19,45], however, only the hCoV-NL63 S used ACE2 for infection (Fig. 4a). While the hCoV-NL63 S mediated efficient infection of cells expressing human, civet, raccoon dog, pig, camel or bat ACE2, it was unable to utilize pangolin and ferret ACE2 (Fig. 4a). Sequence analyses revealed that only the latter ACE2 orthologs contained substitutions (G354R and G354H) in an otherwise highly conserved glycine residue (Fig. 4b). The RBD of the hCoV-NL63 S has been well-characterized[25,45] and residue G354 in ACE2 is directly involved in S-ACE2 interaction (Fig. 4c). Molecular modeling of S/ACE2 interaction using reactive force field simulations confirmed the establishment of close proximity and putative interactions between the RBD of NL63 S and the ACE2 receptors. These analyses predicted that substitutions of G354R and G354H weaken the interaction of ACE2 with the NL63 S protein (Fig. 4d). Altogether, these results indicate that hCoV-NL63 has the potential to infect various animal species and suggest that G354 determines ACE2 usage by this circulating hCoV.

**Sera from SARS-CoV-2 vaccinated individuals neutralize VSVpp infection mediated by S proteins from bat CoVs.** In light of previous coronaviral zoonoses, it is important to clarify whether vaccination against SARS-CoV-2 may protect against future transmissions of bat CoVs to humans. To assess this, we examined sera from ten individuals, five of whom received heterologous ChAdOx1-nCoV-19/2xBNT162b2 and five others who received 3xBNT162b2 prime-boost vaccinations[46,47]. All sera inhibited SARS-CoV-2 VOCs including Omicron BA.5, albeit the latter with reduced efficiency (Fig. 5a). However, they were poorly active against the recombinant XBB.1 Omicron VOC and the XBB.1.5 subvariant containing a S486P mutation in Spike thought enhance viral transmissibility (Fig. 5a)[48–50]. It has been reported that sera obtained after SARS-CoV-2 vaccination neutralize S proteins of SARS-CoV-1 and RaTG13[26,51]. In agreement with this, infection mediated by S proteins of the bat CoV Rs3367, which is closely related to SARS-CoV-1[13], and the SARS-CoV-2-related pangolin CoV GX-P1E[15], were inhibited as efficiently as SARS-CoV-2 variants (Fig. 5a and Supplementary Fig. 8). To test neutralization under particularly challenging conditions, we examined the effects on infection mediated by the BtKY72 S that shows only ~72% amino acid identity to the SARS-CoV-2 Spike. As outlined above, BtKY72 does not use human ACE2 for infection (Fig. 3a). Thus, *Rf* and *Ra* ACE2 were used in these experiments. Despite limited homology to the SARS-CoV-2 S and utilization of bat ACE2, BtKY72 S-mediated viral entry was efficiently neutralized by sera from COVID-19 vaccinated individuals (Fig. 5a and Supplementary Fig. 8). In comparison, only marginal effects were observed on infection mediated by the MERS-CoV S protein that shows only 34.8% amino acid identity to the SARS-CoV-2 S.

The efficiency of neutralization varied between donors in both the AZ2xBNT and 3xBNT groups (Fig. 5b). On average, however, the IC$_{50}$ values of sera from both groups against the different S protein correlated very well (Fig. 5c). The BA.5 S protein was less sensitive to neutralization than Hu-1, with most remaining CoV-2 S proteins showing intermediate phenotypes. The XBB.1 and XBB.1.5 S proteins, however, were largely resistant to neutralization and the IC$_{50}$ was ~tenfold higher than for BA.5 S (Fig. 5b). Sensitivities of S proteins from bat CoVs to neutralization were within the range of Hu-1 to BA.5 S proteins and correlated with their homology to the early SARS-CoV-2 S, with RaTG13 being the most and BtKY72 being the least sensitive (Fig. 5c). In addition, the strength of ACE2 usage had an impact since neutralization of RaTG13 T403R S required higher doses compared to the parental S protein. Similarly, BtKY72 S-mediated entry via *Ra* ACE2 was more efficient than via *Rf* ACE2 (Fig. 3a) and the average IC$_{50}$ for neutralization was two- to threefold increased (Fig. 5c). Altogether, these results suggest that prime-boost vaccination against SARS-CoV-2 may protect against future zoonoses of bat *Sarbecoviruses* but also illustrates the ability of newly emerging SARS-CoV-2 variants to evade humoral immune control.

**Discussion**
The ability of CoVs to cross-species barriers and the importance of utilization of the human ACE2 receptor for successful transmission to humans is established. However, the spectrum of ACE2 receptor usage by animal CoVs that may cause future zoonoses and the mechanisms underlying species-specific differences are poorly understood. Here, we analyzed the ability of S proteins from all seven human CoVs, as well as related CoVs from reservoir bat or potential intermediate animal hosts, to use the ACE2 orthologs from nine different species for viral entry. We found that CoV S proteins differ in their ability to utilize

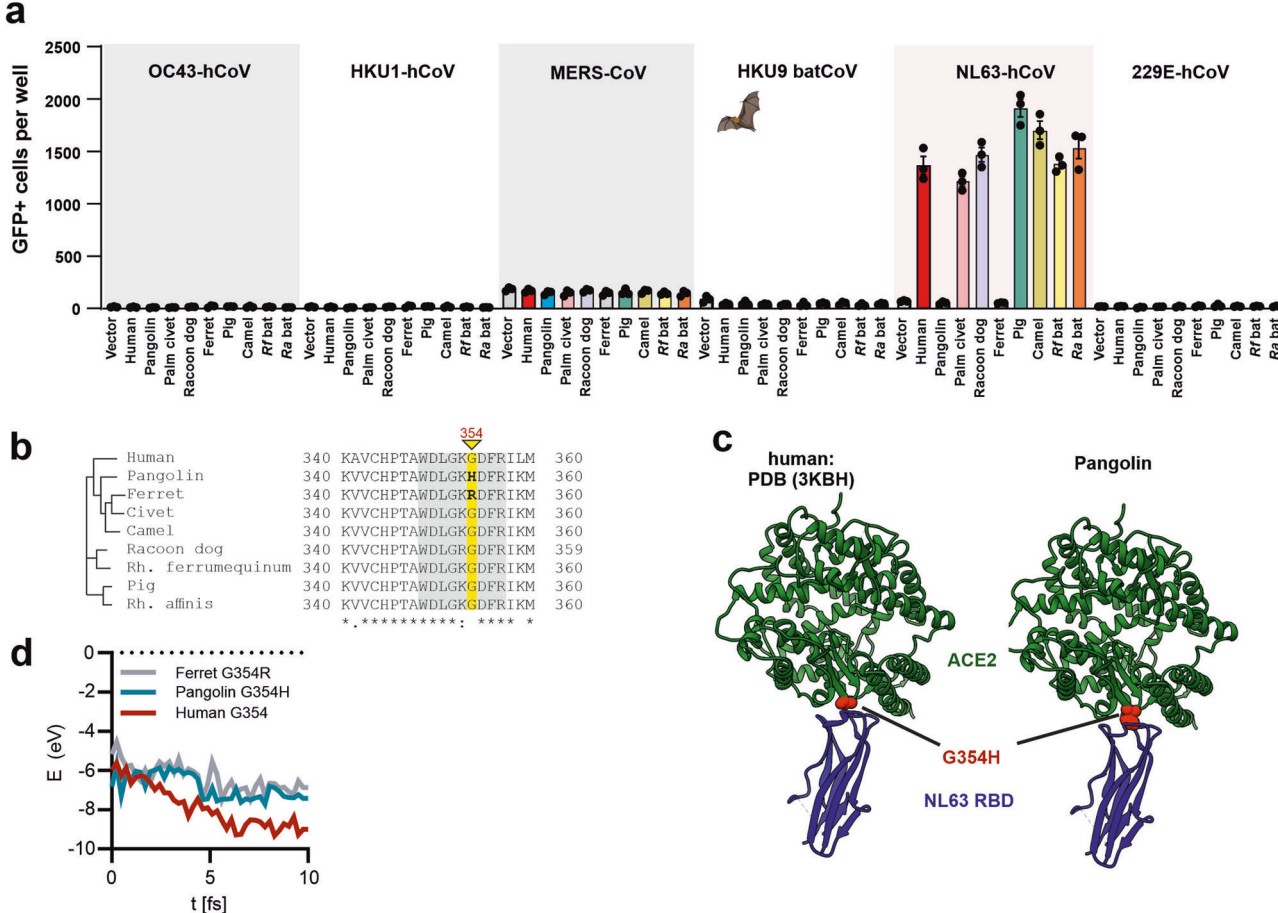

**Fig. 4 Species-specificity of ACE2 receptor usage by hCoV-NL63. a** Infection of HEK293T cells expressing the indicated ACE2 receptors by VSVpp carrying S proteins of MERS-CoV or circulating hCoVs. Bars represent the mean of three experiments (± SEM). **b** Alignment of the amino acids in the corresponding region of ACE2 receptors used for functional analyses. **c** Position of the G354H variation in ACE2 at the interaction site with the RBD of the hCoV-NL63 S protein. **d** Exemplary energy curve of the reactive molecular dynamics simulation for hCoV-NL63 S with human, ferret and pangolin ACE2 receptors.

ACE2 for infection from efficient usage of all orthologs, over species-specific to complete lack of utilization (Fig. 6a). In addition, we provide evidence that changes of R493Q in SARS-CoV-2 and T403R in RaTG13 bat CoV S proteins, as well as D31N, H41Y or G354R/H in ACE2 receptors, play key roles in the species-specificity and efficiency of ACE2 receptor usage by human and animal CoVs. Altogether, our data suggest that ACE2 usage may allow human and animal CoVs to spread between all nine species examined (Fig. 6b). Encouragingly, however, our results further show that sera from individuals vaccinated against SARS-CoV-2 neutralize VSVpp infection mediated by S proteins from divergent bat CoVs via both human or bat ACE2 receptors.

Mutation of R493Q distinguishes BA.4/5, XBB.1 and XBB.1.5 from BA.2. This change represents a reversion to early SARS-CoV-2 strains and enhances S affinity for the human ACE2 receptor and consequently the replicative fitness of BA.4/5 in human cells[52]. Thus, it came as a surprise that mutation of R493Q specifically disrupted the ability of the Omicron BA.2 S protein to use *Rf* ACE2 for infection. These results show that viral adaption for efficient utilization of human ACE2 may come at the cost of losing the ability to utilize the ACE2 receptor from another species. Notably, the early Hu-1 variant also contains 493Q and is unable to utilize *Rf* ACE2 (Fig. 2). Thus, we identified an example of expanded ACE2 tropism from Hu-1 to BA.1 and BA.2 that was lost again in BA.4/5, XBB.1, and XBB.1.5. Published data suggest that Q493R initially emerged because it mediates antibody resistance and then reverted to regain efficient

receptor binding when additional mutations allowed extended antibody evasion[52]. Notably, the effect of the R493Q mutation on ACE2 usage was highly specific since this substitution exclusively impaired the interaction with *Rf* ACE2 and none of the other ACE2 orthologs, including that from *Ra* bats.

We have previously shown that an amino acid change of T403R allows the S protein of RaTG13, one of the closest bat relatives of SARS-CoV-2[10], to efficiently utilize human ACE2 for viral entry[26]. Here, we demonstrate that the T403R change boosts utilization of ACE2 receptors from all species analyzed. In several cases, infection mediated by the parental RaTG13 S was close to background levels but became as least as efficient as infection mediated by SARS-CoV-2 S upon introduction of the T403R mutation (Fig. 3). Notably, the 3- to ~50-fold enhancement of infection by the various ACE2 orthologs in transient transfection assays most likely underestimates the effects of the T403R substitution under more physiological settings. In fact, the enhancing effect of T403R on RaTG13 S-mediated infection of human lung cells or hPSC-derived gut organoids was substantially higher for endogenous ACE2 than in cells overexpressing human ACE2[26]. Our results substantiate the key role of 403R, which is found in the S proteins of most bat CoVs, in efficient utilization of ACE2 from different species.

While T403R RaTG13 S was able to utilize ACE2 from all nine species analyzed, the BtKY72 and Rc-o319 S proteins showed more limited and specific ACE2 usage. As an extreme, the Rc-o319 S only mediated infection via *Ra* ACE2 and only with low efficiency

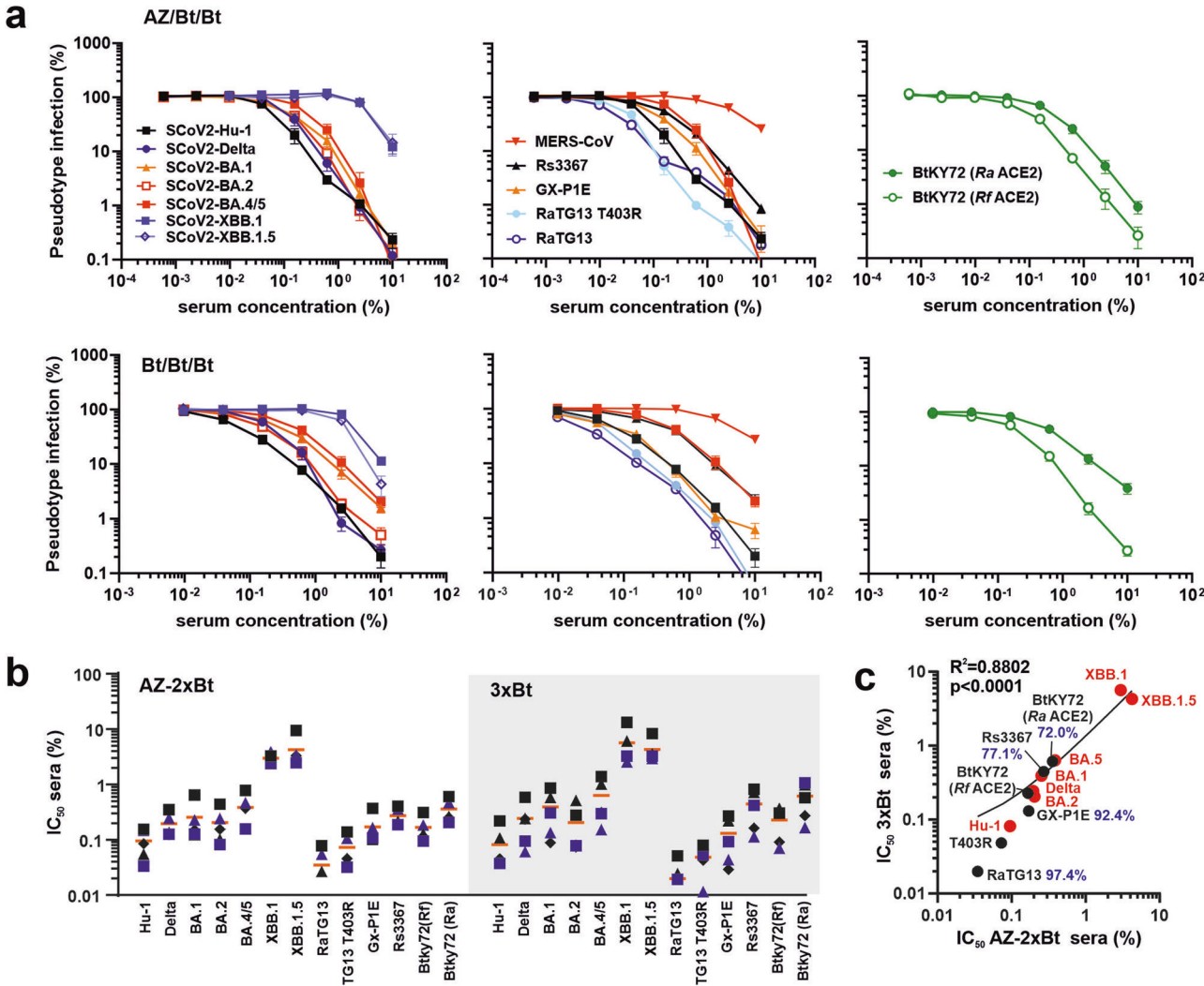

**Fig. 5 Neutralization of S proteins from human and bat CoVs. a** Neutralization of VSVpp carrying the S proteins of the indicated human and bat CoVs by sera obtained from five AZ/BNT/BNT (upper) or five 3xBNT (lower) vaccinated individuals compared to the untreated control (set to 100%). Shown are mean values obtained for the five sera, each tested in at least two technical replicates. Infection was measured in HEK293T cells expressing human or bat (right panel) ACE2. **b** IC50 values obtained for neutralization of the indicated S proteins by individual sera as described under (**a**). Symbols indicate individual donors and orange lines the average value obtained for the five sera from the respective groups. **c** Correlation between the IC50 values obtained for the AZ/2xBt and 3xBt groups. Human S proteins are indicated by red and bat S proteins by black symbols. Percentages indicate amino acid identity to the SARS-CoV-2 S protein.

(Fig. 3a). In comparison, the BtKY72 S used *Rf*, *Ra* and camel ACE2 with high and palm civet ACE2 with moderate efficiency but none of the other five ACE2 orthologs. Our results indicate that mutations of K31N/D/E/T allow utilization of ACE2 orthologs for BtKY72 S-mediated infection. Notably, the geographic distributions and habitats of *Ra* and *Rf* bats overlap[53]. *Ra* is found in many countries in Southeast Asia, while *Rf* is distributed throughout Europe, Asia, and parts of Africa. CoVs closely related to SARS-CoV-2 have been identified in *Ra* bats in China and this bat species is one of the possible reservoir hosts for the origin of the COVID-19 pandemic. Another virus (RmYN02) that is related to SARS-CoV-2 has been identified in *Rhinolophus ferrumequinum* bats in Japan[54]. Our finding that some bat CoVs use the ACE2 receptors from both bat species suggests that SARS-CoV-related viruses may have the potential to spread throughout large parts of Europe, Asia, and parts of Africa by jumping from *Ra* to *Rf* bats. This agrees with the recent identification of SARS-CoV-related *Sarbecovirus*es in European horseshoe bats in Europe[55].

The circulating hCoV-NL63 is also thought to have originated in bats, and closely related CoVs have been detected in *Pipistrellus*

and *Rhinolophus* bats[6,56]. We found that the hCoV-NL63 S efficiently utilizes bat and human, as well as camel, pig, palm civet and raccoon dog ACE2 for infection. Thus, this common cold virus that has been discovered almost 20 years ago[57] should have the potential to infect various animal species. However, to our knowledge, no human-to-animal transmissions have been documented. In contrast, the NL63 S was unable to use pangolin and ferret ACE2 for infection. Lack of function is most likely due to species-specific changes of G354H/R in the ACE2 interaction site with the receptor-binding domain (RBD) of the hCoV-NL63 protein (Fig. 4). Thus, a single amino acid change in ACE2 may disrupt utilization by the S protein of a circulating hCoV and residue G354 seems to play an important role in the species-specificity of S-ACE2 interaction.

One important question is whether COVID-19 vaccination may provide protection against future zoonotic coronaviruses. In support of this, we have previously shown that the T403R RaTG13 S is sensitive to sera from vaccinated individuals[26]. This was expected since RaTG13 and SARS-CoV-2 Spike show ~97.5% identity and sera from vaccinated individuals also neutralize the

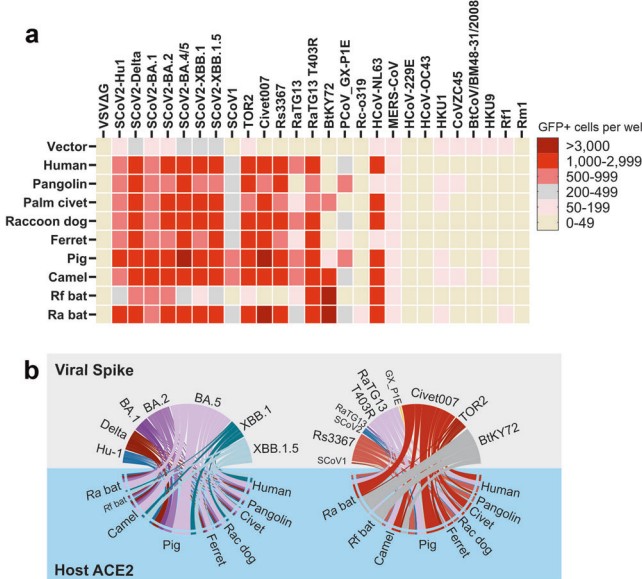

**Fig. 6 Overview on ACE2 usage by CoV S proteins and possible cross-species transmissions. a** Binned heatmap of the infection efficiency of various ACE2-S combination. Data aggregated from Figs. 2a, 3a, and 4a. **b** Chord diagram of the data in (**a**) depicting the efficiency of indicated S-ACE2 mediated infection as scaled connectors. Left panel: SARS-CoV-2 Hu-1 (blue) and Variants of concern (Delta, red; Omicron, purple and green). Right panel: assorted SARS-CoV-2-related CoVs (blue and purple) and SARS-CoV-1-related CoVs (red) compared to bat CoVs (gray). Species of ACE2 as indicated in the lower half of the panel.

SARS-CoV-1 S that shows only 76% homology to that of SARS-CoV-2[58,59]. Here, we expanded these analyses to more divergent bat CoVs. We found that sera from vaccinated individuals even prevent VSVpp infection mediated by the BtKY72 S, which shows ~72.0% identity to the SARS-CoV-2 S, via bat ACE2 receptors (Fig. 5c). On average, S proteins from SARS-CoV-2 variants and different bat CoVs showed similar sensitivities to neutralization, although BA.5 S shows just ~2.7% and the BTKY72 S ~ 28.0% sequence divergence from the Spike proteins used for vaccination. This illustrates that many changes evolving in SARS-CoV-2 Omicron VOCs become fixed because they allow evasion of humoral immune responses.

In conclusion, our results show that ACE2 usage by S proteins of human and animal CoVs is often broad. However, single amino acid changes in both S and ACE2 can drastically change the efficiency of S-mediated virus infection via ACE2 orthologs from specific species. In addition, we show that increased utilization of human ACE2 may come at the cost of losing the ability to use ACE2 from a bat species. Our results further suggest that bat CoVs closely related to SARS-CoVs have the potential to spread between different species of *Rhinolophus* bat that altogether are widely distributed across Asia, Africa and Europe. On a positive note, sera from COVID-19-vaccinated individuals inhibited VSVpp infection mediated by Spike proteins of bat CoVs. While further studies are required to obtain definitive proof, our results suggest that vaccination against SARS-CoV-2 might confer protection against future zoonoses of bat *Sarbecoviruses*.

## Methods
**Cell culture**. All cells were cultured at 37 °C in a 5% $CO_2$ atmosphere. Human embryonic kidney 293T cells purchased from American Type Culture Collection (ATCC: #CRL3216) were cultivated in Dulbecco's Modified Eagle Medium (DMEM,

Gibco) supplemented with 10% (v/v) heat-inactivated fetal bovine serum (FBS, Gibco), 2 mM L-glutamine (PANBiotech), 100 µg/ml streptomycin (PANBiotech) and 100 U/ml penicillin (PANBiotech). Mouse I1-Hybridoma cells (ATCC:#CRL-2700) were cultured in Roswell Park Memorial Institute (RPMI) 1640medium supplemented with 10%(v/v) heat-inactivated fetal bovine serum, 2 mM L-glutamine, 100 mg/ml streptomycin and 100 U/ml penicillin.

**Pseudoparticle production**. To produce pseudotyped VSVΔG-GFP particles, $6 \times 10^6$ HEK293T cells were seeded 18 h before transfection in 10-cm dishes. The cells were transfected with 15 µg of a glycoprotein-expressing vector using TransIT®-LT1 (Mirus). Twenty-four hours post-transfection, the cells were infected with VSVΔG-GFP particles pseudotyped with VSV-G at an MOI of 3. One hour post-infection, the inoculum was removed. VSVΔG-GFP particles were harvested 24 h post-infection. Cell debris was pelleted and removed by centrifugation ($500 \times g$, 4 °C, 5 min). Residual input particles carrying VSV-G were blocked by adding 10% (v/v) of I1-Hybridoma Supernatant (I1, mouse hybridoma supernatant from CRL-2700; ATCC) to the cell culture supernatant.

**Structure modeling**. The structure complex of SARS-CoV-2 BA.2 spike with or without R493Q and *R. ferrumequinum* ACE2 was homology modeled using SWISS-MODEL (https://swissmodel.expasy.org/), based on the structure of SARS-CoV-2 BA.2 spike complexed with mouse ACE2 (mACE2) (PDB codes 8DM7)[60]. The structure complexes of BtKY72 RBD and *R. Affinis* ACE2 with WT/N31D/N31K were homology modeled using SWISS-MODEL (https://swissmodel.expasy.org/), based on the structure of SARS-CoV-2 Hu.1 spike complexed with *R. Affinis* ACE2 (RaACE2) (PDB codes 7XA7)[61]. Molecular graphics visualization and analyses were performed using the UCSF Chimera software (http://www.rbvi.ucsf.edu/chimera).

**Expression constructs**. pCG plasmids coding SARS-CoV-2 Spike Wuhan-hu-1 (NCBI reference Sequence YP_009724390.1), Delta, BA.1, BA.2, BA.5, XBB.1, or XBB.1.5 were kindly provided by Stefan Pöhlmann (Göttingen, Germany). Spike genes are listed in Supplemental Table 1 and were synthesized by Twist Bioscience, PCR amplified, and subcloned into a pCG-strep expression construct using the In-Fusion® HD Cloning Kit (Takara) according to the manufacturer's instructions. pTwist EF1 Alpha-V5 tag plasmids expressing different ACE2 orthologs (supplemental Table 2) were also synthesized by Twist Bioscience. pTwist EF1 Alpha-V5 tag vector, pCG-strep vector, pCG-BatCoV-RaTG13 spike T403R, and pTwist EF1 Alpha-Rhinolophus ferrumequinum ACE2 with mutations of L24R, K27I, D31N, S34H, N38D, H41Y, F83Y S849, or S387T were generated using the In-Fusion® HD Cloning Kit (Takara). SARS-CoV-2 Spike BA.2 spike with mutations of Δ69-70, L452R, F486V, or R493Q were generated using Q5 Site-Directed Mutagenesis Kit (NEB). All constructs were verified by Sanger sequencing in Microsynth seqlab. Primer sequences are listed in Supplementary Data 2.

**GFP-Split fusion assay**. To detect the formation of syncytia, HEK293T cells expressing split GFP1-10 or GFP11 (Kindly provided by Prof. Dr. Oliver Schwarz, the Pasteur Institute, France[62]) were mixed 1:1 at the final density of $6 \times 10^5$ cells/mL. Then 500 µl cells were co-transfected with 350 ng of ACE2 and 350 ng of Spike expressing vectors using LT1. Forty hours post-transfection, fluorescence microscopy images were acquired using

the Cytation 3 microplate reader (BioTek Instruments), and the GFP area was quantified using Fiji ImageJ.

**Cell–cell fusion assay.** HEK293T cells were trans-transfected with Spike-IRES GFP expression plasmids and ACE2 expression plasmids. Forty-eight hours post transfection, the GFP positive cell–cell fusion were recorded by the Cytation 3 microplate reader (BioTek Instruments) and the GFP area was quantified using Fiji ImageJ.

**Whole-cell and cell-free lysates.** Whole-cell lysates were prepared by collecting cells in phosphate-buffered Saline (PBS, Gibco), pelleting ($500 \times g$, 4 °C, 5 min), lysing (1 h, 4 °C), and clearing (14,000 rpm, 4 °C, 10 min). Viral particles were filtered through a 0.45-µm MF-Millipore Filter (Millex) and lysed in 1× Protein Sample Loading Buffer (LI-COR). All protein samples with loading buffer were heated at 95 °C for 10 min and stored at −20 °C.

**SDS-PAGE and immunoblotting.** Whole-cell lysates were separated on NuPAGE 4–12% Bis-Tris Gels (Invitrogen) for 90 min at 120 V and blotted at constant 30 V for 30 min onto 0.45-µm Immobilon-FL PVDF membrane (Merck Millipore). After the transfer, the membrane was blocked in 1% casein in PBS (Thermo Scientific) and stained using primary antibodies directed against strep (1:5000, Thermo Fisher Scientific, PA5-119611), V5 tag (1:1000, Cell Signalling, #13202), VSV-M (1:2000, Absolute Antibody, 23H12, #Ab01404-2.0), or GAPDH (1:1000, BioLegend, #631401). Infrared Dye labeled secondary antibodies IRDye 800CW Goat anti-Mouse #926-32210, IRDye 800CW Goat anti-Rat (#926-32219), IRDye 680CW Goat anti-Rabbit (#925-68071), IRDye 680CW Goat anti-Mouse (#926-68070), IRDye 800CW Goat anti-Rabbit (#926-32211) were used, all 1:10,000. Proteins were detected using a LI-COR Odyssey scanner. Uncropped blots are shown in Supplementary Fig. 9.

**Sera from vaccinated individuals.** Blood samples of ChAdOx1-nCoV-19/ BNT162b2/ BNT162b2 and BNT162b2/ BNT162b2/ BNT162b2 vaccinated non-convalescent individuals were obtained after the participants information and written consent. Samples were collected 11–15 days after the third dose using S-Monovette Serum Gel tubes (Sarstedt). Before use, the serum was heat-treated at (56 °C, 30 min). Ethics approval was given by the Ethics Committee of Ulm University (vote 99/21– FSt/Sta). All ethical regulations relevant to human research participants were followed.

**Sequence logo and alignments.** Alignments of primary viral spike sequences: https://www.ncbi.nlm.nih.gov/nuccore/ (BCN86353.1, QTW89558.1, UFO69279.1, UHU97100.1, UPN16 705.1, QHR63300.2, QIA48632.1, AVP78031.1, AAR86775.1, AG Z48818.1, BCG66627.1, P59594.1, AAU04646.1, ABD75332.1, ABD75323.1, APO40579.1, YP_003858584.1, AVR40344.1, YP_1 73238.1, QBM11748.1, AVP25406.1, YP_003767.1, NP_07355 1.1) and alignments of ACE2 sequences https://www.ncbi.nlm.n ih.gov/nuccore/ (BAB40370.1, XP_017505752.1, XP_01750575 2.1, ABW16956.1, NP_001297119.1, XP_020935033.1, XP_00 6194263.1, XP_032963186.1, QMQ39229.1) were performed in Clustal Omega (https://www.ebi.ac.uk/Tools/msa/clustalo/) using the ClustalW[63] algorithm and an ordered input. The resulting phylogenetic tree was transferred to ITOL (https://itol.embl.de/) and visualized as a rectangular phylogenetic tree in default settings. Sequence logos were generated using R packages ggplot2 and ggseqlogo[64].

**Molecular dynamics simulation.** Molecular dynamics simulations were used to investigate the interaction between the SARS-CoV-2 spike and ACE2. For this purpose, the protein structure from the Protein Data Bank (PDB)[65] with the ID code 3KBH was used. Atomic positions were taken and then equilibrated using ReaxFF simulations[66] within the Amsterdam Modeling Suite 2021 (http://www.scm.com) for 0.5 ns at 300 K. Subsequently, the amino acids in position 354 were replaced and the system was equilibrated again for 0.5 ns at 300 K using ReaxFF simulations. Atomic positions were taken and then equilibrated using ReaxFF simulations within the Amsterdam Modeling Suite 2021 (http://www.scm.com) for 0.5 ns at 300 K, each with at least three independent runs.

**Statistics and reproducibility.** Statistical analyses were performed using GraphPad Prism 9.4.1 (GraphPad Software). $P$ values were determined using a two-tailed Student's $t$ test with Welch's correction. Unless otherwise stated, data are shown as the mean of at least three independent experiments ± SEM.

**Reporting summary.** Further information on research design is available in the Nature Portfolio Reporting Summary linked to this article.

### Data availability
The datasets generated during and/or analyzed during the current study are available from the corresponding authors on request. Source data for figures and blots in the manuscript can be found in Supplementary Data 1. Spike and ACE2 structures, used in this study are available in the Protein Data Bank (PDB) under accession code 3KBH.

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

## Acknowledgements

We thank Regina Burger and Daniela Krnavek for technical assistance. The ACE2 vector and the SARS-CoV-2 S-HA plasmid were kindly provided by Shinji Makino and Stefan Pöhlmann. C.P., A.S., and S.N. are part of the International Graduate School for Molecular Medicine (IGradU). F.Z. was funded by the "Bausteinprogramm", Projektnummer: L.SBN.0225, of Ulm University. This study was supported by DFG grants to F.K. (CRC-1279), T.J. (CRC-1279), and K.M.J.S. (CRC-1279). F.K. and K.M.J.S. were supported by the BMBF (Restrict SARS-CoV-2 and IMMUNOMOD). J.A.M. received funding from the DFG (MU 4485/1-1).

## Author contributions

Q.W. performed most experiments. S.N., Q.X., C.P., and F.Z. supported Q.W. and performed cell-to-cell fusion and neutralization assays. C.J. and T.J. performed molecular modeling analyses. A.S. and J.A.M. provided serum samples. Q.W. F.Z., K.M.J.S., and F.K. conceived the study, planned experiments, and wrote the manuscript. All authors reviewed and approved the manuscript.

## Funding

## Competing interests

The authors declare no competing interests.

**Additional information**

