## [Peer Review File · Communications Biology]

Reviewers' comments:

Reviewer #1 (Remarks to the Author):

Wang et al. conducted an exhaustive study on the interactions between various spike proteins from both human and animal coronaviruses (CoVs) and ACE2 receptors from numerous animal species, utilizing a range of assays. This study yielded several significant findings. Notably, the authors discovered that the SARS-CoV-2 Omicron BA.4/5 binds to various ACE2 receptors more efficiently than its predecessors. In addition, three ACE2 residues were identified as critical for the binding of different coronaviral spikes. Importantly, their results suggest that COVID-19 vaccines can stimulate the production of broad neutralizing antibodies against selected bat Sarbecoviruses. While the insights provided by this study could be invaluable for future pandemic preparedness, I have a few concerns that require the authors' attention:

1. It would be beneficial for the authors to include more current strains of Omicron, such as XBB.1.5 or XBB.1.16, to ensure the timeliness of the study.
2. While the authors used several assays to substantiate their findings, it would be advantageous to incorporate biacore assays. These assays would provide a more precise quantification of the interaction between the spike protein and the ACE2 receptor

Reviewer #2 (Remarks to the Author):

In the paper entitled "Determinants of species-specific utilization of ACE2 by human and animal coronaviruses", the authors report a series of controlled experiments demonstrating determinants of ACE2 receptor usage of diverse CoVs. Although these results were mainly derived from pseudoviruses and have certain limitations, but they were combined with syncytial formation experiments to examine the effect of receptors on virus-induced membrane fusion at the cell fusion level. In summary, the work is well done and comprehensive, and the manuscript could be improved by considering the following minor points:

Line 55. I would add "HCoV-NL63" after SARS-CoV-1.

Line 57. Please change "plays a key role for" to "plays a key role in".

Line 103 and 104. I would change the sentence to "cells expressing pig ACE2 showed the highest infection efficiency".

Line 176. Please specify "this position".

Line 191. Please check for spelling errors and change "variatis" to "variations".

Line 210 and 211. Please change "five who" of the two places to "five of whom", and "five other who", respectively.

Please supplement the method and principles of phylogenetic tree construction in the methods or figure legends to make the results more realistic and convincing.

In addition, the figure format and figure legends may need further optimization. For example, figure 1a might try to shrink the icon to make it look more comfortable. In Figure 3a, we can see the

background traces of pasted species pictures, which also needs to be improved. In figure 4c, the font of the species name corresponding to ACE2 on the left of Figure 4c is obviously different from that in other figures. And in figure 6, please explain "GFU" in the figure legend.

Reply to reviewers' comments (in *italic* letters)

Reviewer #1: Wang et al. conducted an exhaustive study on the interactions between various spike proteins from both human and animal coronaviruses (CoVs) and ACE2 receptors from numerous animal species, utilizing a range of assays. This study yielded several significant findings. Notably, the authors discovered that the SARS-CoV-2 Omicron BA.4/5 binds to various ACE2 receptors more efficiently than its predecessors. In addition, three ACE2 residues were identified as critical for the binding of different coronaviral spikes. Importantly, their results suggest that COVID-19 vaccines can stimulate the production of broad neutralizing antibodies against selected bat Sarbecoviruses. While the insights provided by this study could be invaluable for future pandemic preparedness, I have a few concerns that require the authors' attention:

1. It would be beneficial for the authors to include more current strains of Omicron, such as XBB.1.5 or XBB.1.16, to ensure the timeliness of the study.

We are pleased that reviewers 1 feels that our analyses are “exhaustive” and “yielded several significant findings”. To increase timeliness, we analyzed two additional Omicron strains, i.e. XBB1 and XBB1.5 that currently accounts for most infections around the globe. The new results are shown in the revised Figures 2, 5, 6, S1, S2, S3 and S8 and described and discussed in the revised text (lines 26, 72, 104, 108, 111, 117, 120, 217-219, 234-237, 262 and 270).

2. While the authors used several assays to substantiate their findings, it would be advantageous to incorporate biacore assays. These assays would provide a more precise quantification of the interaction between the spike protein and the ACE2 receptor.

We agree that biacore assays might provide additional useful information. However, this would require purification of the different Spike proteins and be very time consuming. Thus, it is beyond the scope of the present study.

Reviewer #2: In the paper entitled “Determinants of species-specific utilization of ACE2 by human and animal coronaviruses”, the authors report a series of controlled experiments demonstrating determinants of ACE2 receptor usage of diverse CoVs. Although these results were mainly derived from pseudoviruses and have certain limitations, but they were combined with syncytial formation experiments to examine the effect of receptors on virus-induced membrane fusion at the cell fusion level. In summary, the work is well done and comprehensive, and the manuscript could be improved by considering the following minor points:

We thank the reviewer for the positive comments and addressed all suggestions below.

Line 55. I would add “HCoV-NL63” after SARS-CoV-1.

Done (line 56)

Line 57. Please change “plays a key role for” to “plays a key role in”.

Done (Line 58)

Line 103 and 104. I would change the sentence to “cells expressing pig ACE2 showed the highest infection efficiency”.

Done (line 106)

Line 176. Please specify “this position”.

Done (line 180).

Line 191. Please check for spelling errors and change “variatis” to “variations”.

We checked and corrected spelling errors throughout (e.g. line 195).

Line 210 and 211. Please change “five who” of the two places to “five of whom”, and “five other who”, respectively.

Changed as suggested (lines 214-215).

Please supplement the method and principles of phylogenetic tree construction in the methods or figure legends to make the results more realistic and convincing.

We expanded the description of the methods used to generate the phylogenetic trees in the methods (lines 407-418).

In addition, the figure format and figure legends may need further optimization. For example, figure 1a might try to shrink the icon to make it look more comfortable. In Figure 3a, we can see the background traces of pasted species pictures, which also needs to be improved. In figure 4c, the font of the species name corresponding to ACE2 on the left of Figure 4c is obviously different from that in other figures. And in figure 6, please explain "GFU" in the figure legend.

We checked and optimized the figures throughout. In addition, we changed GFU (green fluorescent units) to GFP+ cells per well in Figure 6 for clarity.

REVIEWERS' COMMENTS:

Reviewer #1 (Remarks to the Author):

The authors have satisfactorily addressed all of my concerns.

Reviewer #2 (Remarks to the Author):

I am satisfied with this current revised version.

The reviewers had no remaining concerns and we thanks them for their helpful suggestions.